# Probabilistic TopK Sparse Autoencoder for Interpreting the Activations of Large Language Models

## Abstract

Sparse Autoencoders (SAEs) have emerged as a popular solution for extracting interpretable features from language model activations, enabling mechanistic understanding by decomposing polysemantic neurons into sparsely activated dictionary components. However, existing SAE designs suffer from deterministic, activations that starve gradients to "dead" components, and produce uncalibrated coefficients that provide no meaningful notion of uncertainty. To address these limitations, we introduce Probabilistic TopK SAEs, a novel approach that augments the TopK autoencoder with probabilistic gating through the binary Concrete distribution. This stochastic sampling helps mitigate gradient starvation to dead neurons while producing coefficient magnitudes that are more correlated with the confidence of feature presence. Empirical experiments with GPT-2 and Qwen3 shows that our method achieves consistent Pareto improvements over the baselines in high sparsity settings (small number of activated features) while maintaining a larger set of alive dictionary features. Further, we show that the coefficients magnitude from our approach exhibit stronger correlation between activation strength and interpretability scores, resulting in more faithful explanations for the neurons.

## 1 Introduction

Mechanistic interpretability in NLP aims to understand how language models transform inputs into outputs through their learned representations and circuits (Olah et al., 2020; Bereska & Gavves, 2024). One fundamental challenge is due to *Superposition* (Elhage et al., 2022), namely the fact that language models learn to represent more features than the number of neurons by encoding each feature as a linear combination of neurons, allowing models to compress a large amount of information into a limited parameter space. This compression leads to polysemantic neurons, where individual neurons can respond to multiple different features depending on the context, making direct interpretation of neuron activations extremely challenging.

Sparse Autoencoders (SAEs) (Bricken et al., 2023; Cunningham et al., 2024; Rajamanoharan et al., 2024a; Gao et al., 2025) have recently emerged as a promising approach to this problem. They learn sparse decompositions of model activations using an overcomplete dictionary of features–where the number of dictionary elements greatly exceeds the input dimension–such that each disentangled feature aligns with a single concept (Shu et al., 2025). This allows SAEs to extract a larger set of monosemantic neurons that can be isolated for interpretation through sparsity enforcement. For example, the L1 penalty can be applied in conjunction with the ReLU activation which pushes small coefficients toward zero (Cunningham et al., 2024; Rajamanoharan et al., 2024a), while the TopK activation selects only the K largest pre-activation values and sets all others to zero, guaranteeing exactly K active features per input and allows linear scaling to very large models (Gao et al., 2025). The development of SAEs has enabled significant advancements in mechanistic interpretability, allowing the identification of interpretable features across multiple scales and layers of language models (Templeton et al., 2024; Lieberum et al., 2024) with successful downstream applications in circuit discovery (Marks et al., 2025) and model steering Bayat et al. (2025).

Arguably, one drawback of the previously proposed sparsity-enforcing methods is that they often **produce deterministic, uncalibrated activations**, which are brittle to early "winner-take-all" dy-

namics. The main culprit seems to be that existing sparsity constraints excessively starve gradients of non-selected units (e.g., ReLU, TopK), with features that start slightly less aligned with the data being further suppressed and never recovering. Additionally, while coefficient magnitudes are used to determine feature usage, these values **do not provide any meaningful notion of uncertainty**. This is due to the fact that efficient sizes are scale-dependent and can be arbitrarily rescaled through encoder–decoder weight trade-offs. For example, if a decoder column is multiplied by a constant $\alpha$, the corresponding encoder row can be divided by $\alpha$ with no change to the reconstruction error. As a consequence, the magnitude offer no information that reflects the confidence in feature selection.

To address these two key limitations, in this paper, we propose **Probabilistic TopK SAE**, which augments the standard TopK autoencoders (Makhzani & Frey, 2014) with Binary Concrete gates (Maddison et al., 2017; Louizos et al., 2018). The stochasticity during training helps reduce the invariance to rescaling by introducing an probability distribution over each dictionary component. Moreover, stochastic gating also mitigates the "winner-take-all" collapse by reducing the prevalence of dead units. By annealing the temperature of the Binary Concrete distribution, the model smoothly transitions from exploratory, probabilistic feature usage early in training to sharper, near-deterministic selection at convergence. This yields a larger and more balanced set of alive features while preserving strong reconstruction performance, yielding Pareto improvement in the sparsity-reconstruction frontier.

To demonstrate the effectiveness of our method, we evaluate Probabilistic TopK SAE on the residual stream activations of GPT-2 (Radford et al., 2019) and Qwen3-0.6B (Yang et al., 2025). Across both models, we find our proposal to be a Pareto improvement over the baselines through better reconstruction fidelity at each sparsity level, while providing producing activation magnitudes that are more correlated with the presence of the underlying features. To justify our design, we perform detailed ablation studies over individual model components to understand where the performance gain come from, while performing an exploratory analysis of different temperature values.

In short, our contributions can be summarized in threefold:

1. We introduce the Probabilistic TopK SAE, a modification to the TopK SAE architecture that integrates input-dependent Binary Concrete gates prior to the Top-K mask to enable stochastic feature selection during training (Section 3).

2. We show that Probabilistic SAEs Pareto improve the sparsity and reconstruction fidelity trade-off over the baselines, especially in high sparsity levels (Section 4.2).

3. We demonstrate improved calibration between activation magnitudes and feature presence by evaluating how well features grouped by magnitude bins align with automatic interpretability scores (Section 4.3).

## 2 PRELIMINARIES

In this section, we summarize the key concepts and notations necessary to understand existing Sparse Autoencoders (SAE) architectures and evaluation methods. We follow notation broadly similar to Bricken et al. (2023) and Rajamanoharan et al. (2024a).

Motivated by the *Superposition Hypothesis* (Elhage et al., 2022), SAEs were proposed to sparsely decompose the model's internal activations $x \in \mathbb{R}^n$ as a linear combination of feature directions:

$$\hat{x} = x_0 + \sum_{i=1}^{M} f_i(x) d_i. \tag{1}$$

From Equation 1, the encoder $f(x) \in \mathbb{R}^M$, where $M \gg n$, is a sparse vector of coefficients that encodes the feature presence in the input activation $x$. The reconstructed activation $\hat{x}$ can then be expressed as a linear combination of decoder dictionary of feature directions $d_i \in \mathbb{R}^n$.

### 2.1 RELU SAE

Earlier works (Bricken et al., 2023; Cunningham et al., 2024) used the ReLU activation to ensure that only nonnegative values pass through. The ReLU operation ensures non-negativity and the

L1 penalty on coefficients c encourages sparsity by shrinking small activations toward zero (Equation 2).

$$f(x) = \text{ReLU}(W_{\text{enc}}x - b_{\text{dec}}),$$
$$\hat{x} = W_{\text{dec}}f(x) + b_{\text{dec}}, \tag{2}$$

The SAE learns to accurately reconstruct the activations by minimizing the Mean Squared Error (MSE) between the input $x$ and output $\hat{x}$, where the trade-off between reconstruction accuracy and sparsity can be adjusted through hyperparameter $\lambda_{\text{sparsity}}$, which controls the strength of the sparsity prior (Equation 3).

$$\mathcal{L} = \|x - \hat{x}\|_2^2 + \lambda_{\text{sparsity}}\|a\|_1, \tag{3}$$

## 2.2 TOPK SAEs

More recently, TopK SAEs (Gao et al., 2025) have emerged as an alternative approach that directly controls sparsity using the TopK activation function (Makhzani & Frey, 2014). Rather than relying on soft thresholding via ReLU and L1 regularization, TopK SAEs deterministically select the $K$ largest pre-activation values for each example (Equation 5).

$$a = f(x) = \text{TopK}(W_{\text{enc}}x + b_{\text{enc}}), \tag{4}$$
$$\hat{x} = W_{\text{dec}}a + b_{\text{dec}}. \tag{5}$$

The training objective is simply the reconstruction loss $\mathcal{L} = \|x - \hat{x}\|_2^2$. Due to the precise control over the sparsity level along with the elimination of tuning an L1 coefficient $\lambda$, the TopK SAE has been successfully scaled up to large LLMs with millions of dictionary features while outperforming ReLU autoencoders on the sparsity-reconstruction frontier (Gao et al., 2025). Despite these benefits, TopK SAEs still suffer from fundamental limitations that stem from their deterministic selection mechanism, where gradient starvation prevents dead features from recovering, resulting in a "rich-get-richer" dynamic where early winners monopolize gradient updates.

## 2.3 EVALUATION

Due to the absence of ground truth labels, most studies focus on the intrinsic quality of SAEs based on the intended behaviors (Shu et al., 2025). A natural evaluation assess the reconstruction accuracy and sparsity rate trade-off on a Pareto curve, as these properties are explicitly enforced in the training loss. In particular, reconstruction accuracy can be measured through the Mean Squared Error (MSE) between the input and reconstructed activated, or Explained Variance (Karvonen et al., 2025), which measures how much variance in the original data is retained after the SAE reconstruction. On the other hand, sparsity rate can be measured through the $L_0$ rate by counting the number of nonzero activations per input. Additionally, another metric to consider is the number of alive dictionary features. Since SAEs with high fraction of dead features represent unused model capacity which can otherwise be used to identify rare but important concepts.

Since the primary goal for SAEs is to enhance interpretability by disentangling LLM activations into meaningful features, the interpretability of SAEs features directly measures their quality through applicability. This can be effectively achieved through automatic interpretability scores, where a LLM is used to generate explanations for SAE features based on input snippets that activates the corresponding neurons. The quality of the explanations is then measured by whether it captures the behavior of the neuron on other text snippets. For example, Bills et al. (2023) used GPT-4 to predict the activations of the neuron in a given context given the generated explanation. The interpretation is then scored by how much the simulated activations correlate with the true activations. In our experiments, we adapt the Detection score from a recently proposed evaluation pipeline Paulo et al. (2025), which demonstrated a higher correlation with human judgment in explanation quality.

## 3 PROPOSAL

Our proposal augments the TopK SAE, which has been successfully scaled up to LLMs with millions of dictionary features through a controlled budget, by placing Binary Concrete gates prior to the TopK operation. The overview of our approach is presented in Figure 1.

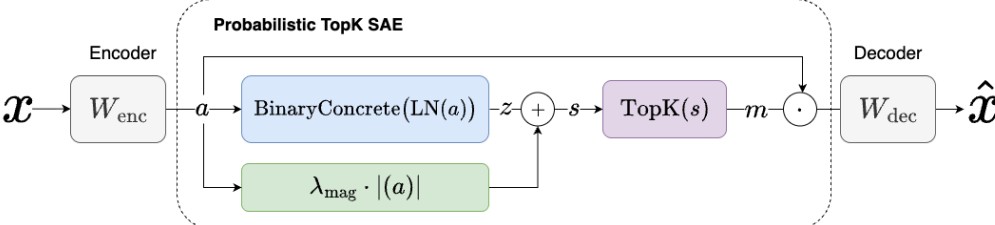

Figure 1: Overview of our proposed Probabilistic TopK SAE architecture. LN denotes the Layer Normalize layer. The samples $z$ from the Binary Concrete gates are combined with the scaled pre-action magnitudes as scores for the TopK selector.

### 3.1 PROBABILISTIC GATING

The binary concrete distribution Maddison et al. (2017) is a continuous relaxation of the Bernoulli random variable. It enables stochastic gating during training by adding Gumbel noise $u \sim \mathrm{Uniform}(0, 1)$ to the log-odds parameter $\log \alpha$, and passing the result through the sigmoid function $\sigma$.

$$p(\alpha) = \sigma\left(\frac{\log \alpha + \log u - \log(1 - u)}{\beta}\right) \tag{6}$$

In Equation 6, $\beta$ is the temperature parameter to control the softness of the approximation. This mechanism allows for the stochastic exploration of binary decisions while remaining amenable to gradient-based optimization. Originally proposed for learning sparse neural networks via discrete parameter selection (Louizos et al., 2018), we adapt the binary concrete distribution for learning input-dependent probabilistic gates to perform context-specific feature selection in sparse dictionary learning.

### 3.2 PROBABILISTIC TOPK SAE

Our proposed model extends the standard TopK Sparse Autoencoder (SAE) by introducing probabilistic gating through the binary concrete distribution. Given an input activation vector $x \in \mathbb{R}^n$, the encoder first computes the pre-activation vectors $a \in \mathbb{R}^M$ by first centering the input with decoder bias $b_{\mathrm{dec}}$. We then apply layer normalization to the pre-activations to stabilize the subsequent gating process by preventing large-magnitude activations from dominating the Binary Concrete sampling (Equation 7).

$$a = W_{\mathrm{enc}}(x - b_{\mathrm{dec}}),$$
$$z = \sigma\left(\frac{\mathrm{LayerNorm}(a) + \log u - \log(1 - u)}{\beta}\right). \tag{7}$$

To reduce the influence of random noise from the stochastic gating, we then compute a combined score $s$ that incorporates both the probabilistic gate samples $z$ and the absolute magnitudes from deterministic pre-activations $a$. Here, we use a hyperparameter $\lambda_{\mathrm{mag}}$ to control the influence from the deterministic magnitude. In practice, we set $\lambda_{\mathrm{mag}}$ to a extremely small value. The combined score $s$ is then used to compute a binary mask $m$ through the TopK operation. Finally, this masked activation is passed through the decoder to produce the final reconstructed activation $\hat{x} \in \mathbb{R}^n$ (Equation 8). During inference, we compute the sigmoid function without Gumbel noise such that $z = \sigma\left(\mathrm{LayerNorm}(a)/\beta\right)$.

$$s = z + \lambda_{\mathrm{mag}}|a|$$
$$\hat{x} = W_{\mathrm{dec}}(m \odot a) + b_{\mathrm{dec}}, \quad \text{where} \quad m_i = \begin{cases} 1 & \text{if } s_i \in \mathrm{TopK(s)} \\ 0 & \text{otherwise} \end{cases} \tag{8}$$

The training objective is the standard mean squared error (MSE) between input and reconstructed activations $\mathcal{L}_{\mathrm{MSE}} = \|x - \hat{x}\|_2^2$. Our formulation mitigates the "winner-take-all" collapse by enabling the model to stochastically explore more feature during training, while enforcing sparsity constraint through TopK gating. The overview of our proposal in presented in Figure 1.

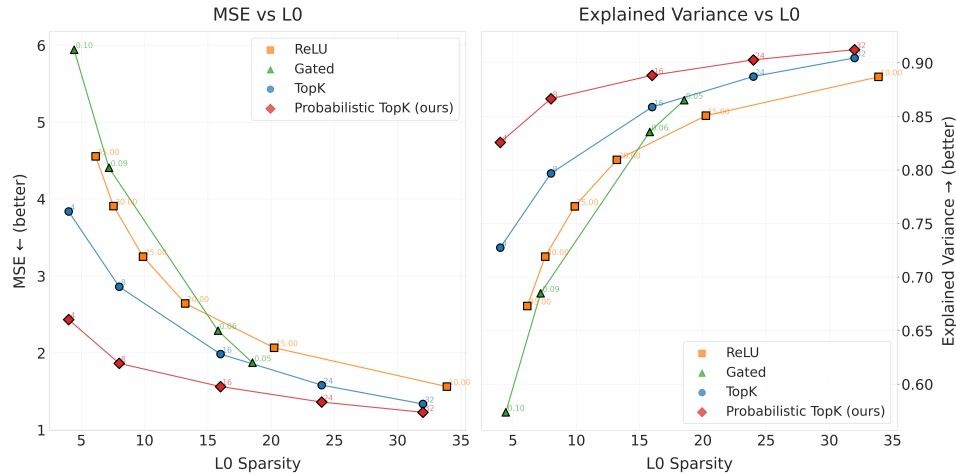

Figure 2: Reconstruction vs. sparsity Pareto curve for layer 8 of GPT-2 small. Probabilistic TopK has a better reconstruction-sparsity trade-off than other activation functions.

### 3.3 ANNEALING SCHEDULE

To balance exploration and stability in feature selection, we anneal the temperature parameter $\beta$ of the binary concrete distribution over the course of training $t \in [0, T]$. Initially, a higher $\beta$ encourages the gates to remain stochastic and exploratory, allowing the model to sample a diverse set of features and avoid premature convergence. As training progresses, $\beta$ is exponentially decayed, making the gates increasingly sharp and binary: $\beta_t = \beta_0 \cdot (\beta_T/\beta_0)^{t/T}$. This gradual transition ensures stable feature selection at convergence.

## 4 EXPERIMENTS

### 4.1 SETTINGS AND BASELINES

Following prior studies (Gao et al., 2025), we perform experiments by applying SAEs on the residual streams of GPT-2 (Radford et al., 2019) and Qwen3-0.6B (Yang et al., 2025). We train SAEs on OpenWebText (Gokaslan & Cohen, 2019) and FineWeb (Penedo et al., 2024) datasets for GPT-2 and Qwen, respectively. All experiments are trained with a context length of 1024. We subtract the mean over the hidden dimension and normalize to all inputs to unit norm, prior to passing to the autoencoder. During training, we gradually anneal the temperature $\beta$ from $5.0$ to $1e^{-4}$ using a exponential decay after the initial warm-up steps. Detailed hyperparameter settings are described in Appendix A.

For comparison, we include three baselines SAE: the standard ReLU SAE described in subsection 2.1 (Bricken et al., 2023; Cunningham et al., 2024), TopK SAE (Gao et al., 2025) (subsection 2.2), and Gated SAE (Rajamanoharan et al., 2024a) that separates the computation of magnitude and activation detection using separate affine transformations. Following Gao et al. (2025), we apply SAEs on a layer near the end of the network, which should contain more meaningful features without being specialized for next-token predictions. Specifically, we use layer 8 for GPT-2, and layer 26 for Qwen3.

### 4.2 PARETO PERFORMANCE

From Figure 2 and Figure 3, we see that our proposed Probabilistic TopK SAE achieves a Pareto improvements over the TopK SAE baselines across all $K$. This is more pronounced in the high sparsity (i.e., $K = 4, 8$) settings, where our model significantly outperforms the baselines in reconstruction accuracy. A key benefit of our design is the use of an initially high temperature parameter ($\beta$), which encourages the model to explore a broader range of dictionary components during early training. This exploratory phase prevents premature commitment to suboptimal feature subsets and

allows the autoencoder to discover a more diverse set of candidate features. As training progresses, $\beta$ is gradually annealed, shifting the model from exploration to exploitation and enabling it to converge on sparse, high-quality representations that preserve reconstruction fidelity even at very low $K$. However, we do see a diminished benefit at lower sparsity level (i.e., $K = 32, 64$), where exploration becomes less important for the model as we allow more selections through the TopK selector, this is especially pronounced at $K = 64$ (Figure 3), where it is outperformed by Gated SAE in reconstruction accuracy.

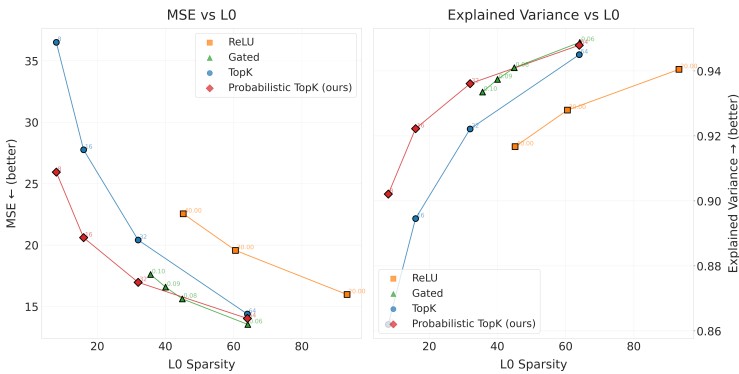

Figure 3: Reconstruction vs sparsity Pareto curve for layer 26 of Qwen3-0.6B. Probabilistic TopK has a better reconstruction-sparsity trade off than Top-K and ReLU, and is comparable to Gated SAE at lower sparsity levels.

We examine the additional benefits of keeping dictionary components alive during training through Figure 4, which highlights the ability of our model to maintain a high number of alive dictionary components during training. If a large number of features switch off during early training, the model's effective capacity is reduced, meaning it has fewer basis functions to represent the data (Bloom, 2024). This can lead to an incomplete representation of the input space and higher reconstruction errors for patterns that lack a dedicated feature. Therefore, maintaining high dictionary utilization (i.e., keeping most features "alive") is crucial for good reconstruction performance: a larger pool of active features enables broader exploration and helps maintain accuracy under the K-sparsity constraint.

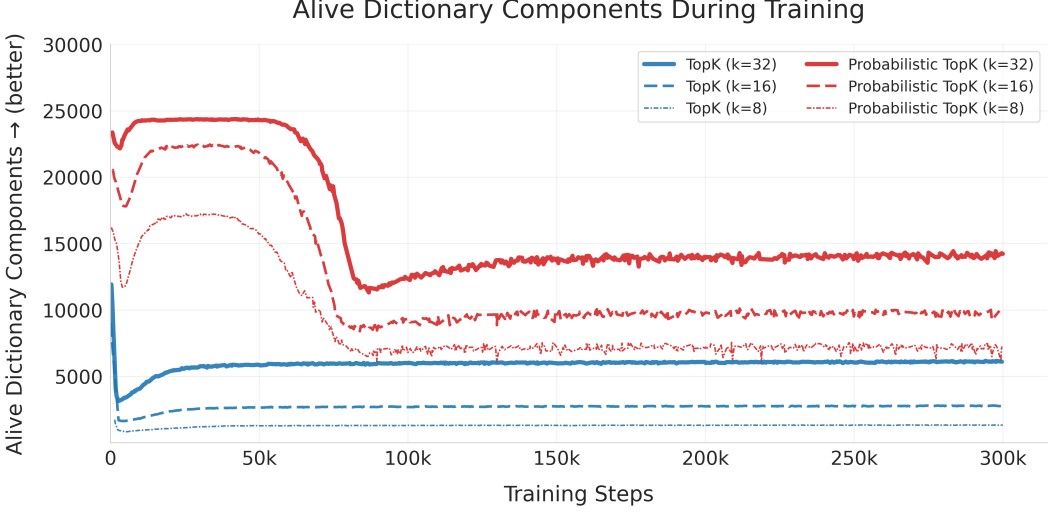

Figure 4: Number of alive dictionary components during training for TopK and Probabilistic TopK SAEs for layer 8 of GPT-2 ($K = 8, 16, 32$). An identification trend is found on layer 26 of the Qwen3 model ($K = 16, 32, 64$) in Figure 8 of Appendix B.

## 4.3 AUTOMATIC INTERPRETABILITY

To evaluate the interpretability of dictionary components and measure the correlation between activation magnitude and interpretability, we perform automatic interpretability using the detection method provided by Paulo et al. (2025), where we stratify neuron activations into percentile buckets and measuring interpretability scores within each stratum. Specifically, we select runs with similar sparsity rate (i.e., $L_0$) and divide each neuron's activation distribution into 5 percentile buckets. We then generate separate explanations from examples in each activation bucket before scoring the explanation from each bucket.

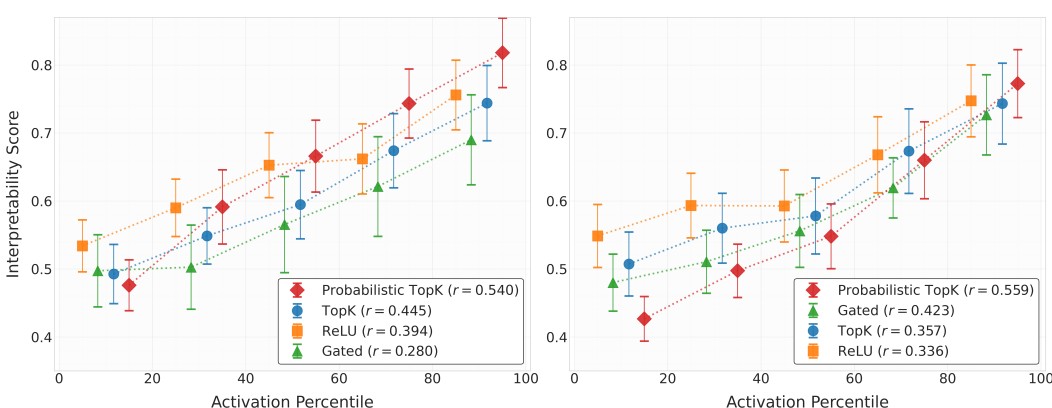

Figure 5: Mean interpretability scores are plotted against activation percentile buckets for four SAE variants trained on GPT-2 small (layer 8). Error bars indicate 95% confidence intervals computed from 150 randomly sampled neurons. We compare methods by choosing runs with similar $L_0$ by including Probabilistic TopK ($K = 8, 16$), baseline TopK ($K = 8, 16$), ReLU ($\lambda_{\text{sparsity}} = 30, 20$, and Gated SAE ($\lambda_{\text{sparsity}} = 0.09, 0.06$). Detailed interpretability scores is presented in Appendix C.

In Figure 5, we present the calibration curves showing interpretability scores as a function of activation percentile, where we find that our Probabilistic TopK achieves highest correlation ($r = 0.540$ for $K = 8$, and $r = 5.559$ for $K = 16$) as well as the highest interpretability score for neurons in the top 20%. The strong calibration of our proposed methods suggests that the probabilistic gating strengthens the relationship between a neuron's activation strength and its confidence to possess the underlying feature. This property allows the better sampling of activations for explanations as well as implications to improved downstream performance such as circuit discovery and model steering.

## 5 ANALYSIS

### 5.1 ABLATION STUDY

To justify our design choices, we conduct comprehensive ablation studies across different sparsity levels ($K = 8, 16, 32$) on GPT-2. We compare our proposed model (Section 3) with the following variants:

- **Top-K**: Described in Section 2.2, standard Top-K SAE without probabilistic gating.

- **w/o Magnitude**: Removes the magnitude from scoring, this is equivalent to setting $\lambda_{\text{mag}} = 0$, such that $s = z$ (Equation 8). This variant eliminates the contribution from deterministic pre-activation values.

- **w/o LayerNorm**: Removes layer normalization layer prior to the binary concrete operation (Equation 7).

- **Sigmoid Gate**: Replaces Binary Concrete with sigmoid gates (Equation 7), this variant eliminates the probabilistic attributes from learning.

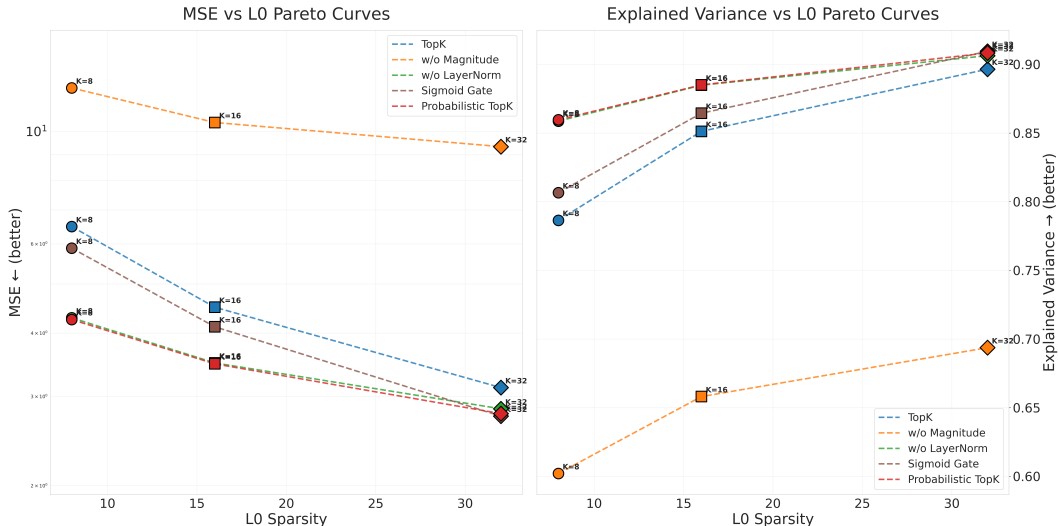

Figure 6: Ablation studies on Layer 8 of GPT-2 ($K = 8, 16, 32$) for the 3 variants include our full model and the TopK SAE baseline without any additional augmentations.

From the results in Figure 6, we see that scoring without magnitude contributions (w/o Magnitude) performs significantly worse than all other variants. While solely using $z$ allow many features to be sampled occasionally, it does not allow the exploitation of small differences in feature alignment with the input, leading to noisy selection and weaker reconstruction quality. Even for very small $\lambda = 1e^{-4}$, this term biases selection enough to select features that are well-aligned with the current input, leading to better reconstruction fidelity. We also find that replacing the Binary Concrete with standard sigmoid (Sigmoid Gate) degrades reconstruction performance in high-sparsity settings ($K = 8, 16$). When $K$ is small, the TopK operator is highly selective where small differences determine which features survive. Since the function lacks stochasticity, it tends to favor a small set of dictionary components, which leads to a suboptimal reconstruction without the exploratory behavior of the Binary Concrete. Lastly, it's worth mentioning that removing layer normalization layer (w/o LayerNorm) does not show significant performance drop, indicating that additional efficiency can be gained by removing the extra parameters associated with the affine transformation.

## 5.2 TEMPERATURE VALUES

Finally, we experiment with different temperatures values to assess its impact on the reconstruction fidelity by running ($\beta \in \{0.5, 1.0, 5.0, 10.0\}$) in GPT-2. From the results illustrated in Figure 7, we find that reconstruction quality is highly sensitive to the temperature parameter, particularly at low sparsity levels. At $K = 8$, increasing $\beta$ from 0.5 to 5.0 reduces MSE by approximately 27% (from 5.83 to 4.25 on layer 10). However, the benefits plateau beyond $\beta = 5.0$, with minimal improvements observed at $\beta = 10.0$. The temperature effect diminishes as sparsity increases. At $K = 32$, the performance gap between $\beta = 0.5$ and $\beta = 5.0$ narrows significantly, suggesting that higher sparsity naturally provides more gradient paths, reducing the importance of exploration through stochastic gating. In high sparsity settings, temperature acts as a regularizer that prevents premature feature specialization.

## 6 RELATED WORK

**SAEs for Language Model Interpretability** Situated within the field of mechanistic interpretability, Sparse Autoencoders (SAEs) encourage localized, sparse activations, yielding interpretable latent features. They have been proposed as a means to explain model behavior across the full training distribution and have been recently applied to language models (Sharkey et al., 2022; Bricken et al., 2023; Cunningham et al., 2024). Following the standard architecture with ReLU activation (Bricken et al., 2023), more recent work has proposed numerous improvements to the original design: refin-

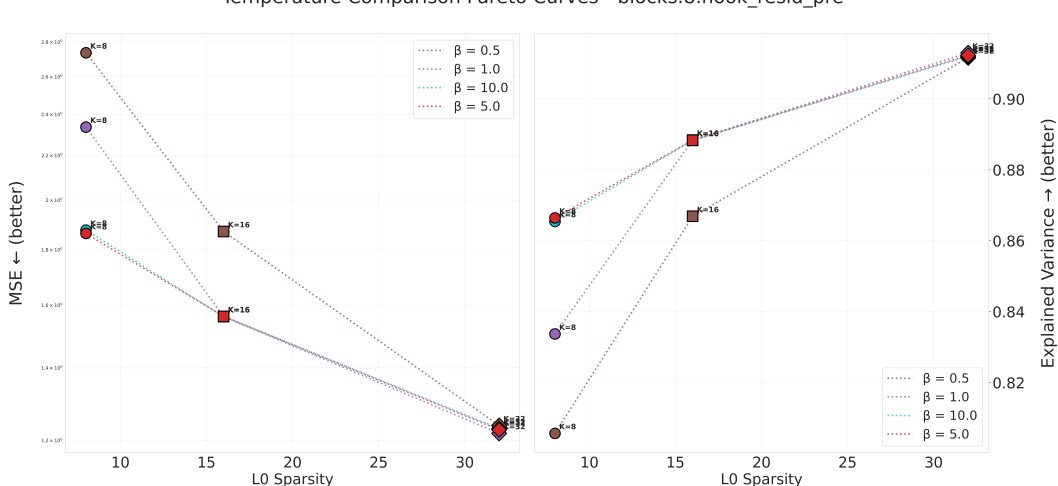

Figure 7: Pareto curve for different temperatures values ($\beta = 0.5, 1, 5, 10$) on Layer 8 of GPT-2 ($K = 8, 16, 32$).

ing architecture (e.g., Gated SAE, Switch SAE, Top-K gating (Rajamanoharan et al., 2024a; Mudide et al., 2024; Gao et al., 2025)), activation functions (e.g., JumpReLU (Rajamanoharan et al., 2024b)), and training objectives (e.g., $\ell_1$ penalties, P-anneal SAE, Feature-Aligned SAEs (Cunningham et al., 2024; Karvonen et al., 2024; Marks et al., 2024)), improving stability, sparsity–reconstruction trade-offs, and faithfulness. Surveys synthesize these trends and discuss evaluation protocols for disentanglement and concept alignment (Shu et al., 2025). Building on this successful line of research, we extend the Top-K SAE (Gao et al., 2025) with probabilistic gating via the binary Concrete distribution to encourage exploratory yet sparse feature selection.

**Stochastic Gating and Relaxed Discreteness**   Introducing stochasticity into selection mechanisms can mitigate premature collapse of feature usage. The binary Concrete distribution (Maddison et al., 2017) and related Gumbel-Softmax relaxations enable gradient-based learning with approximately discrete gates. These relaxations have been used broadly for promoting sparsity, feature selection, and structured pruning (Louizos et al., 2018). Large sparse models similarly combine randomized gating with hard Top-K selection to choose a small subset per input (e.g., noisy Top-K in Mixture-of-Experts and Switch Transformers) (Shazeer et al., 2017; Fedus et al., 2022). Our novel setting adapts this principle to dictionary learning with SAEs: placing binary-Concrete gates before a Top-K operator encourages exploration of alternative feature subsets early in training, while temperature annealing sharpens the gates toward discrete selections at convergence, thus improving coverage and stability relative to purely deterministic Top-K SAEs.

## 7    CONCLUSION

In this work, we introduced the Probabilistic Top-K SAE, an extension of Top-K autoencoders that inserts binary-Concrete (relaxed Bernoulli) gating ahead of the Top-K operator. This stochastic gate encourages input-dependent exploration early in training and, with temperature annealing, sharpens toward discrete selections at convergence. Empirically, the resulting models achieve a stronger Pareto frontier on the sparsity–reconstruction trade-off than deterministic Top-K baselines, reflecting both higher dictionary utilization and greater stability. For future work, we hope to perform additional downstream experiments (e.g., circuit discovery, model steering) to demonstrate the effectiveness of our Probabilistic TopK SAE, as well as improving the existing architecture to achieve better performance at higher $K$.

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

## A    HYPERPARAMETER DETAILS

Using the TransformerLens library (Nanda & Bloom, 2022), we train Probabilistic SAEs on the residual activation directly after the layer norm for both GPT-2 and Qwen3-0.6B models. For all SAEs on both models we use the Adam optimizer (Kingma & Ba, 2015) with a learning rate (lr) of $1e^{-3}$, $\beta_1 = 0.9$ and $\beta_2 = 0.99$. During training, we clip the grad norm above $10.0$, and use a cosine lr scheduler with exponential decay after the initial warm-up steps. We initialize the decoder weights orthogonally and initialize encoder weights as the transpose of decoder weights. For all runs, we set the number of dictionary components to be 16 times the hidden state dimensions. During evaluation, we use a context window of 64 and iterate over 50M tokens. In Table 1, we list all hyperparameters necessary to reproduce Figure 2 and Figure 3. Training for GPT-2 runs on a single A6000 GPU with 48GB of VRAM, while for Qwen, it is done on a single H100 GPU with 80GB of VRAM. All trainings can be completed within 12 hours.

## B    QWEN ALIVE DICTIONARY COMPONENTS

Figure 8 shows an identical trend for Qwen3, where a better dictionary utilization can also be achieved with higher $K$.

## C    AUTOMATIC INTERPRETABILITY SCORES

| Hyperparameter | GPT-2 | Qwen3-0.6 |
|---|---|---|
| Dataset | OpenWebText | FineWeb |
| N samples | 500,000 | 500,000 |
| Total tokens trained | 307M | 215M |
| Learning rate | 5e-4 | 5e-4 |
| Learning rate warmup | 20,000 | 20,000 |
| Learning rate decay | cosine | cosine |
| Batch size | 8 | 8 |
| LLM context length | 1024 | 1024 |
| Layer | 8 | 26 |
| Sparsity coeff (ReLU SAE) | [5, 8, 10, 15, 20, 40] | [10, 20, 40] |
| Sparsity coeff (Gated SAE) | [0.05, 0.06, 0.09, 0.1] | [0.06, 0.08, 0.09, 0.1] |
| K (TopK and Probabilistic TopK) | [4, 8, 16, 24, 32] | [8, 16, 32, 64] |
| Temperature (exponential decay) | [5.0, 1e-4] | [5.0, 1e-4] |

Table 1: SAE training hyperparameters for GPT-2 and Qwen3-0.6.

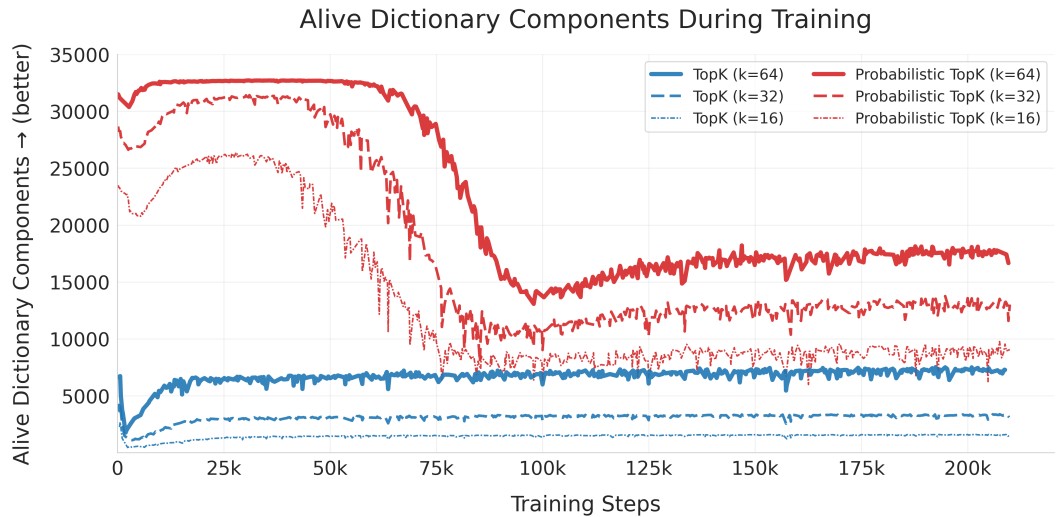

Figure 8: Number of alive dictionary components during training for TopK and Probabilistic TopK SAEs for layer 26 of Qwen3 ($K = 16, 32, 64$).

Table 2: Interpretability Scores by Activation Percentile and Correlation

| Method | 0-20% | 20-40% | 40-60% | 60-80% | 80-100% | Correlation |
|---|---|---|---|---|---|---|
| **Set 1:** $K = 8$, $\lambda = 30/0.09$ | | | | | | |
| ReLU | 0.534 | 0.590 | 0.653 | 0.662 | 0.756 | 0.394 |
| Gated | 0.497 | 0.503 | 0.565 | 0.621 | 0.690 | 0.280 |
| TopK | 0.493 | 0.549 | 0.595 | 0.674 | 0.744 | 0.445 |
| Probabilistic TopK | 0.476 | 0.591 | 0.666 | 0.743 | 0.818 | 0.540 |
| **Set 2:** $K = 16$, $\lambda = 20/0.06$ | | | | | | |
| ReLU | 0.549 | 0.593 | 0.593 | 0.668 | 0.747 | 0.336 |
| Gated | 0.480 | 0.511 | 0.556 | 0.619 | 0.727 | 0.423 |
| TopK | 0.507 | 0.560 | 0.578 | 0.673 | 0.743 | 0.357 |
| Probabilistic TopK | 0.427 | 0.497 | 0.548 | 0.660 | 0.773 | 0.559 |

