# OpenReview forum: "Probabilistic TopK Sparse Autoencoder for Interpreting the Activations of Large Language Models"
_ICLR.cc/2026/Conference — Submitted to ICLR 2026_

### Official Review · Reviewer_pcNN · 2025-10-28

**Soundness:** 2
**Presentation:** 3
**Contribution:** 2
**Rating:** 4
**Confidence:** 4

**Summary:**

The authors introduce Probabilistic TopK SAEs, which use probabilitic gating before applying the TopK operation. They mitigate the phenomenon known as dead neurons, where some latents stop getting gradients during training, and seem to improve on the reconstruction loss when compared with TopK SAEs. They also show that examples that have the highest activations are also more clearly identifiable to a given feature by showing that the correlation between activation and interpretability is higher.

**Strengths:**

The proposed architecture seems to beat a good baseline (TopK SAEs) at reconstruction.

The proposed architecture seems to have a lower amount of dead neurons than regular TopK SAEs.

**Weaknesses:**

Although it seems clear that the proposed architecture is better at a lower L0, the trend seems to indicate that other architectures might win out at higher L0s. Some recent literature developments indicate that low L0s might not be recovering good dictionaries but instead have better reconstruction loss due to the wedging of features. On all the figures this seems to be happening, where the trend seems to indicate a saturation for probabilistic TopK the same is not exactly happening for TopK or Gated SAEs.

The number of dead features for the TopK architecture is extremely high, making it possible that the parameters of these SAEs where badly tuned (very high learning rate for instance).

The current version of this work does not substancially improve on the SAE state of the art, and requires substantial work to merit publication at a conference like ICLR. New architectures should not only improve on the L0 vs MSE pareto curve but show their usefullness or new characteristics that show them to be more interesting than previous architechtures.

**Questions:**

Why do the number of alive neurons start so much lower for TopK SAEs? How are you counting dead neurons?

Why does the number of alive neuron not start at the same number for the different K?

How much slower is this new architecture to train?

---

### Official Review · Reviewer_oYEB · 2025-10-28

**Soundness:** 2
**Presentation:** 3
**Contribution:** 1
**Rating:** 2
**Confidence:** 3

**Summary:**

SAEs are often used for extracting interpretable features from LM activations.
The authors identify two problems with traditional SAEs: firstly there can be dead features and once features die they don't get any gradient signal to be revived and secondly there is no measure of the uncertainty around a given feature.
They introduce Probabilistic TopK SAEs (PSAEs) which adds a probabilistic gating function to the SAE using a Concrete distribution. They claim that this also converts the feature magnitudes into more clearly defining a confidence of feature presence.

They show increased performance considering the MSE and Explained Variance Pareto plots however seem to show reduced performance in the (automated) interpretability evaluation which makes it difficult to see if their approach is an improvement over prior SAE architectures. It is also difficult to see if the baselines that the authors use are strongly tuned which slightly weakens the confidence in their results.

Overall this was a well written paper with a clear goal and a neat, well-motivated approach. Clarifying some of the questions below especially about their evaluations and the baselines would make this a stronger paper.

**Strengths:**

- Clearly identifies two potential problems with previous SAEs and looks to fix those problems
- Figure 1 helps in explaining how the method works
- The authors used Pareto charts to compare their method with other SAE methods and show impressive performance
    - Their outperformance is particularly striking in the very high sparsity (low k) range
    - (For high sparsity however their approach seems to be equivalent or perhaps worse than other baselines)
- Similarly the automated interpretability evaluation is useful to include
- Future work of circuit discovery and model steering sounds useful
- Good clarity of exposition
- There seems to be more alive features than the TopK method (though still quite a few dead features in absolute terms)

**Weaknesses:**

- In Section 2.2, the authors suggest that TopK SAEs suffer from "gradient starvation prevent[ing] dead features from recovering"
    - Given the auxiliary loss introduced in this paper to mitigate dead features and their other experiments and ablations on reducing dead features, this section feels like it somewhat misrepresents that work (see Section 2.4 of Gao et al 2024)
- Nit: Could the authors rename the Proposal section to "Methods"?
- Could the authors provide a comparison to the BatchTopK architecture? (Bussman et al 2024). This is a state of the art SAE architecture which is not compared against
- For a more holistic evaluation, using some of the evaluations in SAE-Bench would be a valuable way to demonstrate that the architecture outperforms on downstream tasks
- It appears that the authors method underperforms on the automated interpretability benchmark relative to the other SAE architectures (for k=16, for all activation percentiles; for k=8 for 2-3 out of 5 activation percentiles)
    - Given that the purpose of SAEs is to extract interpretable features, this seems to be a good judge of their method (especially as they increase complexity of the SAE architecture which could overfit to the MSE evaluation) - do the authors have any hypothesis for why PSAEs seem to not work as well for producing interpretable features?
- A natural baseline for their approach seems to be a Gumbel-noised Top-K on the SAE latents without their additional architectural changes, did the authors compare against this and if so what was the comparison?
- The evaluations are on very small models (it is not clear which GPT-2 model but it seems like the 125M param model and the Qwen model is 0.6B params)
    - It would be useful to understand whether the approach scales well (even slightly larger models with a scaling law would be useful if the authors lack compute for very large models)
- It would be useful for the authors to provide an algorithm as well as code for reproducibility
    - In particular, understanding how the gradients flow through with the sampling objective would be valuable for clarifying the approach presented
    - It would also be interesting to hear if there was any instability noticed throughout training with the stochasticity

**Questions:**

- Is aligning feature magnitudes with confidence of feature presence the right objective? It's currently unclear whether feature magnitudes are best thought of as probabilistic confidence in feature or the intensity of the feature - it seems like your approach takes a side on this so I would be interested in hearing more about the motivation
- There are many other methods for reducing the number of dead features from resampling (Bricken et al 2023) to auxiliary losses (Gao et al 2024 and Ayonrinde 2024). Ayonrinde 2024's approach (Feature Choice SAEs) suggests that they get ~0 dead features.
    - How does your approach compare to the previous methods for reducing the number of dead features?
- Given that the ablations show that the LayerNorm does not provide much value but adds complexity, why do the authors keep the layer norm in their architecture?
    - As they note efficiency can be gained without the LayerNorm
- How many dead features were there in the baseline methods?
    - What attempts to remove dead features did you use in the baselines? E.g. resampling, ghost gradients, auxiliary losses etc.
    - Given that the paper is about reducing dead features primarily it would be useful to ensure that the baselines are strongly tuned for this use case
- Similarly, if the baselines do have dead features then for 50% dead features, we should compare the PSAE with a half width to the full width baseline SAE (because the dead features can effectively be pruned from the baselines to get a parameter and FLOP-matched baseline)
    - Could the authors confirm the SAE widths used for the baselines and the proportion of dead features in each?
    - The authors may find the MDL-SAE's description length metric valuable in comparing SAEs of different widths or with different numbers of dead features (Ayonrinde et al 2024). This would likely be a good evaluation metric to add to the Pareto charts.
- What is the definition of feature aliveness used in this work?
    - How sensitive are the results to the aliveness threshold?
- What is the distribution of features that are used by the SAE after training in the PSAE and TopK SAE cases?
    - How much are the dead features used in practise and how different are they from the existing features?
    - One thing we could imagine is that the newly alive features are merely feature splits (Bricken et al) of the features that the TopK SAE learned and are not providing genuine value
    - In general alive features should be a proxy for increased performance (e.g. on an interpretability study) rather than an end in itself and so much of the value of understanding feature aliveness will be in gaining intuition for why such an approach works and where it could be further improved
- Similarly, for the alive but low frequency features how interpretable are these features (by a human or with the automated methodology)?
    - We would like to know that the newly alive features that your method recovers are generally useful for interpretation rather than being noise/polysemantic features which contribute to reducing MSE
- Is the very high sparsity case (e.g. k=4) a typical case for SAEs used by practitioners? Most papers seem to use k in the 20-150 range which seems to be where the PSAE method is less strong
- I would be interested in hearing more details on the relationship between confidence and calibration to AutoInterp
    - It's not obvious that this is the same underlying quality as correlation with interpretability score is not the same as calibrated probability of a concept being present.
    - Since inference is presumably deterministic, the confidence here should be an internal phenomena as the confidence will generally not be surfaced to the consumer of the SAE explanations (whether a human or an AI agent)
    - It would be useful to hear more about how you imagine this working in practise
    - Did you try any stricter notion of calibration, e.g. Brier scores, to test whether z is predicting “feature present” as a probability rather than just that the stronger signal tends to be more interpretable (which is what the automated interpretability score plot reveals)?

---

### Official Review · Reviewer_xABC · 2025-10-29

**Soundness:** 3
**Presentation:** 3
**Contribution:** 3
**Rating:** 6
**Confidence:** 4

**Summary:**

This paper introduces Probabilistic TopK SAEs, which inserts a Binary Concrete Gate in the TopK SAEs, to involve uncertainty in TopK SAE training. This approach solves the problem that traditional sparsity-enforcing methods often produce deterministic, uncalibrated activations, making a harmful winner-take-all training dynamics. Experiments on GPT-2 and Qwen-0.6B shows the pareto improvements on reconstruction versus sparsity, as well as autointerp scores.

**Strengths:**

1. The paper is well-written. Figures are clear. Motivation of methodology and experiments is clarified and easy to follow.
2. Probabilitistic TopK SAEs show substantial improvements in sparse dictionary learning. In training SAEs on GPT-2 and Qwen-0.6B, Probabilistic TopK SAEs improve the pareto frontier of explained variance and L0 norm by non-trivial values. The proportion of alive features also show consistent improvements. These prove the involvement of stochastic terms in SAE training as an effective method in sparse dictionary learning.
3. This stochasticity makes the activation magnitude meaningful: higher activation maginitude suggests higher confidence to posses the underlying feature. This is proven by the high correlation between interpretability scores and activation magnitudes.
4. The ablation studies are complet and inspiring, showing the irreplaceability of magnitude and binary concrete gate.

**Weaknesses:**

1. Lack of generalizability: all experiments are restricted to standard SAEs. More SAE variants could be involved, including transcoders, crosscoders or CLTs, as activation function should be a general method for all kinds of sparse dictionary learning.
2. Lack of scalability: experiments are restricted to GPT-2 and Qwen-0.6B, which is relatively small. It'll be better to involve larger sized models to prove the scalability of the proposed method.

**Questions:**

1. Eq. (8) use $m \odot a$ as the multiplier of $W_\text{dec}$. Since elements in the pre-activation $a$ could be negative, does this mean the feature activation magnitude could be negative, which is a bit unintuitive?
2. Does the stochastic term exist in inference time? Does this mean it will produce different features in different inference results, which is especially unnatural as an interpretability method?

---

### Official Review · Reviewer_89ZY · 2025-10-31

**Soundness:** 2
**Presentation:** 3
**Contribution:** 3
**Rating:** 4
**Confidence:** 4

**Summary:**

This paper introduces a variant of the Sparse Autoencoder (SAE) architecture, called a Probabilistic TopK Sparse Autoencoder. This architecture builds on the TopK SAE by adding a controllable level of noise to the decision of which features are in the TopK. This noise acts as a form of regularization, and seeks to prevent the model from accumulating dead features. The authors demonstrate in Section 4.2 that their architecture produces more accurate reconstructions at a fixed sparsity level than baseline TopK SAEs, and that it has fewer dead features. In Section 4.3, the authors also use autointerpretability scores as evidence that their architecture's features become more interpretable as they increase in activation quantile.

**Strengths:**

The authors have good communication throughout, and do a good job detailing their architecture and training processes.

The results in Section 4.2 are good metrics on which to evaluate their sparse autoencoders, are clearly presented, and show a significant improvement of their approach over other methods (though see the Weaknesses section below).

The authors ablation study in Section 5 is very welcome, and does a good job identifying which factors of their approach are most important to their improvements.

**Weaknesses:**

The primary weakness of this paper is that it ignores the field's existing best practice for dealing with dead neurons, an auxiliary loss function which pushes dead neurons to reduce the model's reconstruction error (sometimes called a "ghost grad"). See Section 2.4 and Section A.2 of (Gao et al, 2025) for the popularization of the auxiliary loss term on TopK-based SAEs, and (Bussmann et al, 2024), (Bussman et al, 2025), (O'Neill et al, 2024), and the implementation of (Gao et al, 2025) in (EleutherAI, 2024) for further examples. The auxiliary loss term greatly reduces the number of dead latents, for instance down to 7% in (Gao et al, 2025), whereas the proposed architecture appears to still have ~50% dead features across several k values (estimated from Figure 4). This paper is not ready for publication until it places its result in the context of TopK SAEs that are trained with an auxiliary loss function. The authors should include it in their discussion of architectures, highlight how the proposed architecture is an improvement over it, discuss whether it is possible to use both techniques, and ideally perform additional experiments as in Section 4.2 to baseline their results against such SAEs.

The results in Section 4.3 have several limitations. First, the results shown here are not the standard practice of reporting overall autointerpretability scores, instead choosing to disaggregate into different activation percentiles and touting a higher correlation between activation percentile and correlation. Second, due to the noisy nature of autointerpretability scores, there is rarely a clean separation between the proposed architecture's scores, and the scores of classical methods. Finally, even taking the mean scores at face value, the authors only find that the novel architecture's features are more interpretable at some levels of sparsity and when comparing relatively high levels of activations. It is not clear whether it is a good thing for low activations of features to be harder to interpret.

**Questions:**

**Questions**

1. Have you assessed if this architecture has an impact on feature splitting (see e.g., Chanin et al, 2025) for background). To the extent feature splitting is "overfitting" on data, and Probabilistic TopK SAEs perform regularization of the data, it could be the case that this architecture is more resistant to feature splitting.

2. Is this technique compatible with BatchTopK (Bussmann et al, 2024), and if so what modifications would be needed?

3. Would it be possible to create a variant of Figures 4 and 8 that show the number of dead latents instead of the number of alive latents, and include those variants in an appendix?

4. Did you experiment with alternative ways of adding noise to the TopK mask? For example, you could add Normally distributed errors to the encoder layer for the purposes of choosing the TopK activations. Or if not, is there a theoretical reason you'd expect Binary Concrete Sampling to outperform it?

5. Did you run your autointerp experiment on the K=32 autoencoders? If so, can you include a figure of those results in the appendix?

6. Did you find overall autointerpretability scores for your architecture compared to other methods (without separating by activation level)? If so, what were they?

7. Do you plan on making your training and evaluation code publicly available after review?

**Comments**

The paper has the following small typos, which the authors may want to fix:

- Line 15-16: There should be no comma in "deterministic, activations"

- Line 203-204: "In practice, we set λmag to a extremely small value" would benefit from a parenthetical statement of what value you use. For instance, you could include "(usually 1e-4)".

- Line 356: There is a typo where you have written "r=5.559", it should be r=0.559 per Figure 5.

- Line 403: Please write \lambda_{mag} instead of \lambda to be more consistent with other sections of the paper and to remove ambiguity with the other instances of unadorned \lambda.

- Line 129: In "an L1 coefficient λ", including the subscript _{sparsity} can reduce ambiguity.

- Line 352: In the sentence "Detailed interpretability scores is presented in Appendix C", "is" should be "are".

**References**

(Gao et al, 2025) Scaling and evaluating sparse autoencoders. https://arxiv.org/pdf/2406.04093

(Bussmann et al, 2024) BatchTopK Sparse Autoencoders. https://arxiv.org/pdf/2412.06410

(Chanin et al, 2025) A is for Absorption: Studying Feature Splitting and Absorption in Sparse Autoencoders. https://openreview.net/forum?id=LC2KxRwC3n

(Bussman et al, 2025) Learning Multi-Level Features with Matryoshka Sparse Autoencoders. https://arxiv.org/pdf/2503.17547

(O'Neill et al, 2024) Disentangling Dense Embeddings with Sparse Autoencoders. https://arxiv.org/pdf/2408.00657

(Cunningham et al, 2023) Sparse Autoencoders Find Highly Interpretable Features in Language Models. https://arxiv.org/pdf/2309.08600

(EleutherAI, 2024) eai-sparsify Library https://github.com/EleutherAI/sparsify

---

### Meta-Review · Area_Chair_J7d7 · 2026-01-05

**Summary:**

Reviewers pointed to poor evaluations (relying on L0 and naive autointerp) and possibly undertuned baseline SAEs. Multiple reviewers pointed out a lack of engagement with standard practices for dealing with dead neurons.

**Reviewer Concerns:**

No rebuttal N/A

**Reviewer Scores:**

No rebuttal, same scores.

---

### Decision · Program_Chairs · 2026-01-26

Reject